# The Enabling Role of Digital Technologies in Sustainability Accounting: Findings from Norwegian Manufacturing Companies

Olena Klymenko [ID], Lise Lillebrygfjeld Halse *[ID] and Bjørn Jæger [ID]

Faculty of Logistics, Molde University College—Specialized University in Logistics, P.O. Box 2110,
NO-6402 Molde, Norway; olena.klymenko@himolde.no (O.K.); bjorn.jager@himolde.no (B.J.)
* Correspondence: lise.l.halse@himolde.no

**Abstract:** Sustainability accounting is an emerging research area receiving growing awareness. This study examines the role of digital technology in manufacturing companies' sustainability accounting. To guide the research, we use a triple layered business model canvas, which supports the accounting of a manufacturer's performance for the economic, environmental, and social aspects of sustainability. We present an explorative case study of four Norwegian manufacturing companies representing different industries. The findings from the study indicate that while accounting for economic values is well taken care of, companies do not perform comprehensive environmental and social accounting. Furthermore, we observed a shift from a focus on sustainability issues related to the internal manufacturing process to a focus on sustainability issues for the life cycle of the product. Even though the manufacturers are at the forefront with regard to automation and control of production, with extensive use of robots giving a large amount of data, these data are not utilized towards sustainability accounting, showing that sustainability and digitalization are seen as two separate phenomena. This study sheds light on how digital data available from applied Industry 4.0 technologies could enhance sustainability accounting with limited efforts, linking sustainability and digitalization. The results provide insights for manufacturers and researchers in moving towards more sustainable operations and products.

**Keywords:** sustainability accounting; manufacturing; digitalization; triple layered business model canvas

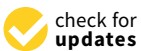



## 1. Introduction

Sustainability and digitalization are two terms that have gained increased attention since they represent potential transforming forces of businesses and society. Sustainability is a radical transformation toward a sustainable society [1,2] and has moved from being a regulative pressure from the surroundings and a corporate buzzword to becoming a concept that businesses have to relate to and implement in their activities. The growing focus toward sustainability is evident in the UN Sustainability Agenda [3], followed by national and international laws and regulations such as the EU Green Deal [4]. Furthermore, in addition to financial accounting and management accounting, businesses start to incorporate environmental and social sustainability variables, which is supported by a variety of approaches—the Global Reporting Initiatives (GRI) [5], ISO certificates [6], internal control regulations [7], and regulations for corporate social responsibility reporting in accounting [8].

Manufacturing companies now need to respond to the demand for sustainability by the market and society at large. Manufacturing industries are here referred to as industries that use highly equipped machines and digital instruments that are helpful in the industries' production. These industries work with large machinery, digital and complex mechanical instruments, drills and cranes, and other heavy transport equipment and appliances [9]. It is crucial for these industries to handle waste that can have hazardous effects on the environment. The manufacturing sector is a cornerstone of the economy

and is crucial to sustainable economic growth but is bound by a tradition where change is slow and costly [10,11]. Furthermore, manufacturers must make decisions that incorporate sustainability factors at strategic, tactical, and operational levels [12]. To accommodate for this, manufacturers experience an increased need for reliable data for (a) external reporting and (b) internal decision-making in their move toward more sustainable operations and products.

Accounting for sustainability has been receiving growing attention in the literature [12–14], supported by new requirements and regulations that will be introduced in the future [4]. While financial accounting is a well-established system of businesses, there is a lack of research on accounting across environmental and social dimensions of sustainability [15]. Environmental accounting includes various indicators from energy, materials, water, waste, and emissions, while societal accounting incorporates multiple effects for stakeholders and local communities [16]. The fourth industrial revolution enhanced innovation in materials and manufacturing, creating a new environment for technology development for data collection and processing [17,18]. The digitalization of manufacturing operations can provide data and information for sustainability accounting systems, facilitating decision-making and enabling more sustainable business processes [19]. Industry 4.0 is considered a framework that can facilitate companies achieving sustainability goals. However, although a range of studies has recently been conducted to investigate the role of Industry 4.0 in enabling sustainable operations [20,21], they are not directed to the topic of sustainability accounting. Hence, it is necessary to clarify the meaning of digital technologies in accounting for environmental and societal parameters. Tiwari and Khan [22] indicate that there is minimal empirical evidence on the adoption of Industry 4.0 enabling technologies for supporting sustainability accounting. Furthermore, Burritt and Christ [12] highlight the need to investigate the Industry 4.0 potential for environmental accounting.

This paper responds to these needs by presenting an explorative case study investigating how manufacturers approach sustainability accounting. The research is guided by the following research questions (RQs)—RQ1: How are manufacturing companies accounting for economic, environmental, and social values? RQ2: To what extent do they apply digital technologies to support economic, environmental, and social accounting? The RQs are addressed through in-depth case studies of four companies using a triple layered business model canvas (TLBMC) [2] to guide the investigation.

The organization of the rest of the paper is as follows. Section 2 presents the background and rationale for sustainability accounting as established by other researchers. Section 3 presents the research method and case study; followed by a discussion in Section 4; and conclusions, limitations, and directions for future research in Section 5.

## 2. Literature

### 2.1. Sustainability and Sustainability Accounting

Sustainability is frequently referred to as "development that meets the needs of the present without compromising the ability to future generations to meet their own needs" [23]. An overwhelming amount of information indicates that we are now living in the Anthropocene—an era in which "human actions have become the main driver of global environmental change" [24]. To move toward more sustainable operations and make the necessary changes, we must be able to account for the effects of the problematic aspects of human actions. Despite the long-lasting attention paid to sustainability, the move toward a sustainable society is slow, proving that the transformation into sustainable operations is a challenging task. This has prompted the emergence of social and environmental accounting focusing on "the impact organizations have on society and the ecology" [25].

In the 1970s, the first wave of corporate sustainability in the form of social reports from Western companies emerged, but later lost its momentum [26]. In the late 1980s, however, reporting on environmental issues gained increased awareness [27]. Since then, the literature has grown substantially, and with increasing attention to sustainability, reporting social and environmental dimensions of corporate activities has become common [26].

Beggington and Larrinaga [25] claim, however, that external sustainable reporting has often had little to do with sustainable development, indicating a decoupling between reporting and the realities. In other words, external sustainable development reporting may be more or less consciously used to portray organizations as being more concerned about sustainability while instead running business as usual [28]. Similarly, accounting literature has lost its connection with social and ecological concerns. Consequently, Bebbington and Larrinaga call for a (re)envisaging of the intellectual roots of accounting for sustainable development so that sustainable development accounting is more in line with sustainable development thinking [25].

Tiwari and Khan [22] describe sustainable accounting and reporting as a "framework for defining sustainability variables based on the triple bottom line model (TBLM), defining and implementing measurement techniques, and reporting the actual status of the variables in the public reports by a company" ([22] p. 1). The sustainability accounting and reporting framework includes standards in line with the TBLM. However, reliable and valid measurement approaches of these variables have been difficult to establish in the industries [12,29]. Burritt and Christ [12] report that companies may lack the technology for collecting timely and appropriate data, which could undermine the credibility of environmental efforts and open channels for greenwashing accusations [30–32]. Burritt and Christ [12] claim that to meet the "environmental imperative of the future" ([12] p. 27), companies need better information technology and richer information.

While financial accounting has evolved over several centuries, hence providing a detailed view of companies' financial operations, accounting systems for environmental operations and social values are, however, still under development [33].

The data collected through sustainable accounting may have several purposes. By collecting these data, companies present their status and maturity regarding environmental and social values to external actors [34]. Based on sustainable accounting data, internal decisions on operations can be optimized to focus on sustainability goals—maintenance, re-use, and remanufacturing—and recycling goals—reduction of material consumption, greenhouse gas emissions, and waste [12,35]. Furthermore, the data provide important input in a company's strategic process, thus enabling decision-making toward sustainable business models [36–38].

The point of departure of this study is the fundamental need for organizations to change to more sustainable operations [39,40]. Sustainability accounting and associated tools help organizations to assess their impact on the environment in which they operate [26]. Even though sustainability accounting tools have been present for several decades, there is, according to Burritt and Christ [12], "a lack of appropriate data, or the technology to collect appropriate data" ([12] p. 25). They mention several avenues for future research, which sums up to aiming at establishing "how Industry 4.0 might facilitate more accurate, high quality, real time environmental management accounting and external environmental reporting in relevant sectors, company sizes, across different management roles and collaborative settings, as well as in intraorganizational settings such as supply chains" ([12] p. 34).

Responding to this call for research, this study aims to explore how manufacturing companies that have moved to Industry 4.0 are applying digital technologies for sustainability accounting. In this study, we focus on the TLBMC [2] to guide our investigations. The TLBMC is mainly a tool for designing more sustainable business models, which is one of the purposes of sustainable accounting data. However, the data collected may provide valuable information for two other purposes. The TLBMC provides an understanding of how organizations generate impacts and how to evaluate these impacts in terms of the triple bottom line perspective. Moreover, the TLBMC integrates the evaluation of impacts throughout the supply chain to customers and stakeholders.

We subsequently present the TLBMC framework in more detail before proceeding to address how digital technologies may play a role in sustainability accounting.

## 2.2. The Triple Layered Business Model Canvas (TLBMC)

The literature suggests a large number of frameworks and methodologies for sustainability, such as sustainable and circular business models [2,41,42]. Sustainable business models integrate economic, environmental, and social values together with interorganizational networks that are broader societal systems [43]. Sustainable business models allow for enhanced understanding of the connection between environmental and social activities and the economic result of the company, thus stimulating managers to balance the activities within these three dimensions. Joyce and Paquin [2] suggested the TLBMC in 2016, aiming to address the environmental, social, and economic dimensions of sustainability through the lens of the business model canvas [41]. The TLBMC is one of the business models for sustainability and consists of (1) an economic layer based on the original business model canvas [41]; (2) an environmental layer that reflects the approach of the life cycle assessment (LCA); and (3) a social layer that is linked to the stakeholder view.

The TLBMC is a visualization of how a company generates economic, environmental, and social values. Thus, at the bottom of each layer, there is a result assessing how revenues outweigh costs for the economic layer, environmental benefits and environmental impacts for the environmental layer, and the social surplus based on the impacts and benefits achieved. Furthermore, the model has horizontal and vertical coherences, meaning that environmental and social practices are interconnected with each other as well as with the economic performance [2]. The model with three layers is presented in Figure 1.

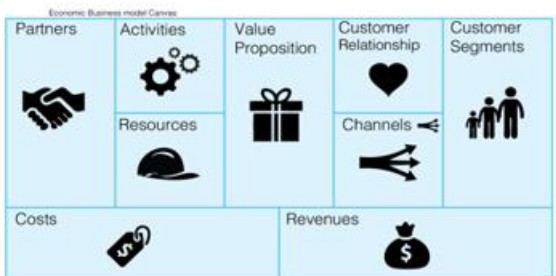 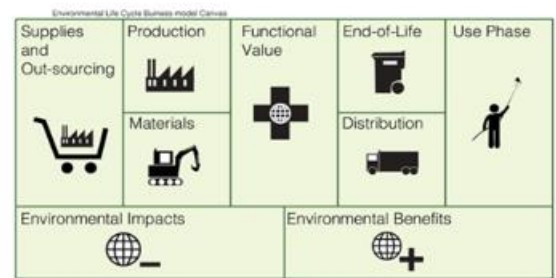

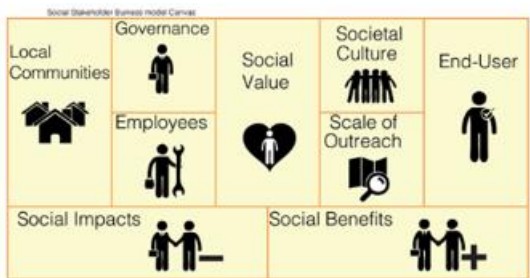

**Figure 1.** The triple layered business model canvas (Joyce and Paquine, 2016).

The economic layer represents the original Osterwalder business model canvas that consists of the following components: customer value proposition, segments, customer relationship, channels, key resources, key activities, partners, cost, and revenues [41]. The environmental layer of the TLBMC does not apply the entire approach of the LCA, but it provides a lens of life cycle thinking, starting from the extraction of raw materials to the end of the life cycle of a product. The social layer represents the social impact of a company on internal and external stakeholders such as employees, suppliers, the local community, the government, and NGOs. As the manufacturers investigated in this paper did not have any knowledge of the TLBMC model and its parameters, we used the TLBMC to guide our investigations on a general level without quantifying specific variables at each layer.

### 2.3. Digitalization and Industry 4.0

Digitalization is receiving growing attention in supply chains among both practitioners and academics. New technology transforms data and information flows and may increase the efficiency of production operations. Its most characteristic feature beyond previous mechanical and electrical technologies is its inherent property of generating data related to the digitalization of tasks or processes. This can be used to analyze the process and find issues for improvement that, when implemented, can generate more data that over time can create a dynamic environment, typically resulting in the entire process being transformed into a new process, as described in the seminal work by Zuboff in 1988 [42]. This ability to generate data from the operations being digitized represents a significant potential for supervising all digitized operations. Digitalization is the driver of the term Industry 4.0 concerning digitalization of manufacturing industries. Digitalization is transforming both manufacturers and their associated global value chains [44,45] into digital manufacturing systems that enable communication between machines and digitized (smart) products [46,47]. Interestingly, the large amount of data resulting from the digitalization of operations and products have the potential to facilitate sustainability accounting at virtually zero cost since the data already exist. This research seeks to find to what extent manufacturers are utilizing this potential by answering the posed RQs developed in the Introduction.

### 2.4. Digitalization and Sustainability

Industry 4.0 brings unique opportunities for environmentally sustainable manufacturing [48]. However, there is lack of studies addressing the intersection between digitalization and sustainability. Digitalization has been identified as an enabler of sustainability in terms of improving resource efficiency and manufacturing performance by the flexible and smart use of digital data [19]. Strandhagen et al. [49] elaborate on how the digitalization of manufacturing logistics influences business operations and sustainability, and give an overview of how trends in logistics relate to each other and to sustainability. Digital technologies provide the information needed to create iterative and restorative systems, thus enabling the companies to move toward sustainable operations and products. Moreover, the study explores how emerging technologies can improve the sustainability of logistics operations in existing business models and how these technologies can contribute to developing new and more sustainable business models. By coupling Internet of Things (IoT)-enabled innovation with sustainability principles, companies may identify new business models. Strandhagen et al. offer a model that can be used as a framework for analyzing businesses and value chains and their interplay between logistics, business models, and sustainability. The model is conceptual, based on literature and foresight studies [49]. Through digitalization, it is possible for manufacturers to improve and optimize capacity utilization, resource efficiency, and inventory management operations, and apply technology for predictive maintenance [19]. Moreover, digital technology can facilitate decision-making during uncertainty [50].

Based on a literature review and expert interviews, [21] qualitatively assessed "the potential of industrial value creation in Industry 4.0 in terms of its contribution to the shift toward sustainable value creation for sustainable value creation" ([21] p. 255). The point of the departure for this study was the United Nations' 17 Sustainable Development Goals (SDGs) and the characteristics of Industry 4.0. The study assessed the macro and micro potential with suggestions of how Industry 4.0-related technologies could facilitate sustainability.

The above-mentioned examples on how technology under the Industry 4.0 umbrella may enable sustainability does not, however, explicitly address sustainable accounting. Digital technologies have the potential for detailed accounting across the economic, social, and environmental layers. Coupling digital technologies with sustainability principles can help companies acquire more accurate information on their operations.

Burritt and Christ [12] claim that the focus on the development of Industry 4.0 has been on "reduced errors, improved product quality, freeing humans from menial and/or

dangerous tasks and providing customers with the products they desire at times when they desire them" ([12] p. 24). They acknowledge that resource efficiency has been emphasized and claim that the understanding of how a broader concept of corporate sustainability could be incorporated into Industry 4.0 is underdeveloped. To address this issue, they examine how corporate sustainability—through environmental accounting—might be incorporated into developing a vision for Industry 4.0. They mention four potential improvements in external environmental accounting ([12] p. 30):

- Better data quality (timeliness, accuracy, reliability, comparability)
- Reduced opportunity for "greenwash" and "brownwash"
- Less management discretion regarding measurements
- Higher credibility of data

The paper provides examples on how these potential improvements may contribute to improve sustainable accounting, but it does not provide any empirical evidence of this.

The role of environmental management accounting is seen as a framework for internal decision-making [51], since well-designed environmental management accounting (EMA) systems may facilitate the environmental and financial aims of the companies. Despite a tendency to apply EMA for short-term operational decisions, it is expected that long-run investment decisions will benefit from EMA evaluation tools. One of the findings in the study by Burritt et al. [51] is the incremental rather than radical changes of EMA practices in case companies. The authors suggest future research to investigate how and when EMA supports and drives more radical changes in organizations toward sustainable development.

A recent study by Tiwari and Khan [22] explores "what technology and architectural features of Industry 4.0 can enable reliable and valid measurements of sustainability accounting variables in the triple bottom line model?", and "how can industries use these features in practice to ensure reliable and valid measurements of sustainability accounting per the GRI framework?" ([22] p. 2). Here, the GRI provides a universal standard for triple bottom line variables. The study comprises primary data from two focus groups and five interviews carried out in companies in small-scale industries in India, where the respondents were asked how existing Industry 4.0 solutions (in general and offered by Indian vendors) could contribute to sustainability reporting. The findings from this study revealed that the sustainable variables were only partially covered in the focus group discussion, and that the interview respondents were cautioned against extreme optimism in investing in Industry 4.0 due to the low maturity level of artificial intelligence (AI). The authors of the paper comment that the results of the study may be affected by the context in which the study was carried out and are characterized by barriers such as low investment in technology and limited competence in sustainability reporting and technology.

Burritt and Christ [12] mention several avenues for future research, which sums up to aiming to establish "how Industry 4.0 might facilitate more accurate, high quality, real time environmental management accounting and external environmental reporting in relevant sectors, company sizes, across different management roles and collaborative settings, as well as in intraorganizational settings such as supply chains" ([12] p. 34).

## 3. Research Method

The RQs call for an explorative approach that can be applied in an in-depth case study combined with secondary data. The case study research method provides an opportunity to explore themes where we have limited knowledge. Yin [52] calls the case study method an empirical inquiry, which examines a phenomenon (the case) in depth and within its real-world context and when the boundaries between the phenomenon and the context are not clear. The rationale for applying a case study is its ability to draw on broader perspectives of the phenomenon under study, thus making it possible to understand a real-world case [52]. Thus, case studies carried out in manufacturing companies can shed light on the sustainability accounting practices and the role of the digital technologies. We have selected a multiple-case design that allows for a broader insight of the study phenomenon.

The first stage of the research addresses how companies conduct sustainability accounting. The second stage explores how digitalization technologies are applied in sustainability accounting. To guide our research, we used the TLBMC [2].

*Case Companies and Data Collection*

Eisenhardt [53] points out that sample size is crucial for achieving research results and defining limits for the generalization of the findings. Consequently, to ensure the external validity of the findings on how the companies account for operations within environmental and social sustainability layers, this study applied a multiple case study analysis. The companies were chosen based on their orientation toward sustainability and digitalization. The case firms for this study represent Norwegian manufacturing companies from four different sectors as presented in Table 1.

**Table 1.** Case companies description.

| Company | | Revenue NOK, Mill | Number of Employees |
| --- | --- | --- | --- |
| Company A | Pipe manufacturer | 1079 | 276 |
| Company B | Furniture producer | 236 | 112 |
| Company C | Maritime equipment manufacturer | 1095 | 347 |
| Company D | Plastic products manufacturer | 74 | 31 |

The information gathered during the interviews was supplemented by secondary data collected from public annual and sustainability reports, website contents, relevant articles from local newspapers, laws, and regulations. A total of eleven interviews were conducted between 2018 and 2021. Data collection started in 2018 when an interview on examining TLBCM was conducted with two Norwegian companies. Later, the idea for the research was expanded to investigate sustainability accounting and the role of digital technology. Hence, the TLBCM contributed to a large extent at the beginning of the research project and was applied as a framework throughout the research. While the first interviews were conducted in a face-to-face setting, the last round of interviews were carried out using the Teams conferences software due to COVID-19 restrictions. Each interview lasted for approximately 1–2 h. The interview questionnaire had a semi-structural design, which comprised an identified set of questions and an opportunity to ask additional questions during the interview process. This allowed for adjusting and reviewing the phenomenon extensively, applying additional information. All interviews were taped and transcribed. To obtain a structured data analysis, a qualitative data analysis computer software package NVivo was used to classify and sort the information necessary for a comprehensive analysis of findings.

## 4. Results

### 4.1. Company A, Pipe Manufacturer

Company A is a Norwegian subsidiary of an international manufacturer of plastic piping systems for heating and plumbing, water pressure, electricity, cable ducting, gas, and agricultural sectors. A large amount of the company's production is allocated to exports. In Norway, the company has three plants and its own research and development department that continuously invests in new products to provide sustainable and environmentally friendly production in Norway.

### 4.1.1. Company A's Sustainability Accounting and Reporting

Sustainability is part of the company's strategy that began with an environmental protection initiative in the mid-1980s. Since 2010, the company has implemented environmental accounting to demonstrate its social responsibility long before it became more common for organizations to improve and share their sustainability performance. The company currently has the UN's 17 SDGs framework as a base for their evaluation of

current performance and identification of goals. Hence, the company's sustainability policy states that to be sustainable, the company needs to systematically address the SDGs by applying internal company goals for the Norwegian department; external company goals with respect to suppliers, customers, and the market; and personal goals related to each employee.

> *"We have two concepts that we have extensively worked on for decades. We had already stated in the 1980s that it is appropriate to address environmental protection. Then we started with EPD, LCA, and third-party certified documentation, and we realized that with the available materials and properties, we can document it."* (Respondent)

Since 2010, Company A has carried out environmental accounting with a focus on $CO_2$ emissions according to the Greenhouse Gas Protocol [54] in two of their manufacturing facilities. The main purpose of the accounting is to demonstrate to customers how the company is working on the reduction of their carbon footprint. Furthermore, the company performs emissions accounting not only to reduce their carbon footprint but also with regard to six additional environmental and climate parameters. Among the motives for implementing the emissions accounting system, the director of sustainability also describes it as a way to demonstrate the general sustainability profile of the company to potential customers:

> *"In future, we will address two types of accounting: environmental and economic accounting. With focus on sustainability, we can achieve positive results for both. At the same time, we can leave a sustainable planet for future generations."* (Respondent)

Almost every year, the management team updates environmental measures and goals. The company recently developed its first Environmental Product Declaration (EPD) that presents footprint per specific product during all phases of a product's lifespan. It is based on the standardized requirements declared in ISO 14,025 Environmental Labels and Declaration Type III and applied to four product groups. Moreover, the company holds ISO 9001 and ISO 14,001 certificates. The company must also comply with the European Regulation, Evaluation, Authorization, and Restriction on Chemicals (REACH) directive. The company states that by providing an EPD, it contributes, in a broader context, to the calculation of the total footprint per building or infrastructure project. Moreover, it provides the opportunity to compare the sustainability of materials and products. Although the company is not obliged to apply for footprint approval in accordance with the Pollution Control Act, it works internally toward the reduction of environmental output within the following aspects: use of resources such as energy and water; choice of raw materials and processes; material balance; and footprint in the air, water, Earth—mainly waste—and transportation of goods. Company A considers environmental sustainability as an avenue for future economic growth as sustainable innovations will open more market opportunities, particularly through the digitalization of pipe systems.

Within the social layer of sustainability accounting, parameters are not so well defined compared to the environmental layer, in which international standards such as ISO 14,001 exist. The company selected the UN's SDGs framework for socially related goals, targets, and indicators. Hence, the management works on prioritizing 3–4 goals out of 17 and concurrently working toward balancing the general recommendations for social sustainability goals. However, as the interviewee stated, it is more reasonable to work toward improvements within environmental aspects. The managers concurrently stated that in future, they expect the social sustainability aspect to become more comprehensive, thus requiring reporting on detailed data and parameters.

The established requirements and regulations for compliance and reporting for sustainability practices often put pressure on and motivate supply chain actors to improve performance and reporting [55–57]. Although this company mainly uses environmental accounting for internal purposes, there are some national regulations in the market through which it operates. The Norwegian government decided to implement environmental requirements for public procurement, which is seen as a driving force for "green transition."

*"Until now, there has been no requirement that has asked about CO$_2$ emissions and EPD, so we have not clarified our climate accounts, but it is something that if we do it, we want to show it. Having said that, the EU Green Deal is under implementation . . . "* (Respondent)

However, Company A's management recently realized that some of the public organizations have not implemented the regulations and do not consider environmental and climate related requirements in public procurement.

### 4.1.2. The Role of Digital Technologies in Company A

The respondent in Company A stated that information technology is seen as part of the current transformation toward a more sustainable and digitalized supply chain. The company is using digital technologies in manufacturing, and sensors are installed to monitor pipe systems at the end-customer's location. In production, the company focuses on digital technologies for interconnectivity and the improvement of effectiveness as sensors are installed to measure energy and water consumption. When the company bought a new product line, they first purchased and installed Industrial Internet of Things (IIoT) technology in the form of sensors to connect all the product lines with an artificial intelligence system. Through machine learning, managers work on optimizing product line processes, energy effectiveness in production, reduction and use of material, and the general optimization of a product's profile. Moreover, the application of machine learning is now considered for quality control in the pipes' automation line. The company along with a partner in telecommunications have developed a new digital monitoring system for pipes and sump systems. With the help of small sensors mounted inside the pipes combined with Narrowband (NB) IoT technology, the information on water conditions, temperature, water level, and other data are collected and sent to a Cloud platform. This remote monitoring control provides an opportunity for the customer to have an early identification of problems.

*"Combined with lifting the sensor data up in a Cloud platform, applying algorithms, machine learning, artificial intelligence, and everything that is possible now, preferably combined with data on weather conditions, we can predict events before they happen"* [58]

The company considers technology development and digitalization to be driving forces for more sustainable buildings and facilities, which can stimulate the reduction of greenhouse gas emission. This provides new economic opportunities to reduce costs and improve the competitiveness of the Norwegian business. One of the departments is currently working on development of IIoT and Response to Intervention; thus, the company can offer additional services such as data analyses and machine learning based on real–time monitoring and data provided through IIoT. Increasing the use of 3D printing provides an opportunity for product development and design that can reduce defects in manufacturing.

The company is currently working on a project for digitalizing their value chain, allowing the storage of all data in product information modeling and the building of information modeling databases including all physical, mechanical, hydraulic, and environmental data. For instance, construction companies that have installed products from the company can easily access complete data regarding the whole building as well as data on a specific component through digitalization.

### 4.2. Company B, Furniture Producer

Company B is one of the largest furniture producers in Norway and is known for its internationally recognizable furniture brands. The company was founded in 1934 in north-west Norway. The company has ten manufacturing locations in the USA, Lithuania, Thailand, and Vietnam, and five factories based in Norway. They provide products in large parts of the world through their own sales companies or via importers.

### 4.2.1. Sustainability Accounting and Reporting at Company B

An essential factor of the company's sustainability strategy has been the growing customers' requirements that can be divided into retailers and end users. A respondent stated that the customer requirements have fundamentally changed compared to five years ago. Consequently, it was important for the company to revise its sustainability strategy.

> *"We have customers with their own end-user perspective where it is about sustainable materials and how they see a sustainable product; it also concerns the entire value chain and everyone around us."* (Respondent)

The new strategy will broadly focus on and emphasize the life cycle of the product, while the previous version mainly focused on manufacturing operations. The company publishes annual sustainability reports as part of the legal requirements for business reporting specified in the Norwegian Accounting Act and the statement of corporate social responsibility [8]. The company reports its greenhouse gas emissions in three scopes: the first includes direct emissions from internal transport and heating with natural gas, oil, and carbon dioxide during foam production; the second comprises indirect emissions from the generation of electricity by the electricity provider; and the third involves emissions from the treatment of waste, air travel, and authorized business use of motor vehicles. Thus, transport is identified as a significant contributor of emissions as goods are transported globally. Even though the company itself aims to collect data to report these parameters, it has a contract with a consultancy firm that performs the data analyses necessary for annual sustainability reports. This may indicate limited internal competence to process and interpret sustainability data. Concurrently, the consultancy firm may not be able to identify possible reasons for deviations and changes in parameters compared to previous periods. The respondent stated that they experience that, compared to smaller companies, Company B must comply with a comprehensive set of regulations and requirements.

The company aims to make environmental information available through EPDs as well as providing objective and open information about how the company handles its environmental responsibility. Company B follows the requirements for strength, stability, and security by the Norwegian Møbelfakta [59]. It is also involved in the UN's Global Compact initiative [60]. Furthermore, the company monitors energy use, which currently relies on 95% of hydropower electricity generation in Norway, on fossil fuel oil that was planned to be replaced by the end of 2020, and woodchips—a by-product of the manufacturing process—which is a main energy source at the Norwegian factories. The company is also continuously working on the reduction of waste by sorting. It is using by-products and recycling where possible, and non-recyclable waste is being utilized for energy-recovering processes such as heat and electricity production. The company acts in accordance with ISO 9001:2015 and ISO 14001:2015 standards. Social aspects are provided as part of their annual sustainability reporting. In the reporting of social sustainability, the company mainly focuses on the internal policy and practices directed toward employees—for example, reporting on gender equality, activities directed to follow the Anti-Discrimination and Accessibility Act, health and industrial safety, personal development, and the apprenticeship program.

### 4.2.2. The Role of Digital Technologies at Company B

Overall, the technology and digitalization level in Company B can be described as relatively high. Thirty years ago, the company began its first efforts on automatization. The company has automated and digitalized processes such as the manufacturing of wooden components, painting, and sewing. Robotization and automation make production processes more effective, contributing to meeting the demand and building competence to remain competitive in terms of production in a high-cost environment.

> *"New technologies, environment-friendly materials, and new product solutions have resulted in one of the most efficient manufacturing environments in the furniture industry today."* (Respondent)

By introducing new means of technology for various operations, the company has digitalized most of the product development process where they are using 3D printing in the prototyping process. Furthermore, designers are working on a joint digitalized platform where the rest of the employees can work on their own tasks, which makes it easier to introduce changes in the product development process and reduce the lead-time.

Even though digitalization plays an important role in the production and product development phases, the company lacks the direct application of technology for the measurement of environmental and social values. However, the company claims that through digital technology, they can receive relevant data for sustainability reporting and decision-making:

> *"Obtaining relevant data and—not least—the opportunity to make good analyses across different data sources is becoming more and more important in order to be able to make the right decisions"* [61]

Based on the aim for a broader focus on sustainability taking the value chain perspective, it is expected that technology will be considered as a tool for realizing these plans in the future, especially for tracking product information along the life cycle and tracking supplier information and their practices.

### *4.3. Company C, Maritime Equipment Manufacturer*

Company C is a mechanical equipment manufacturer that supplies products for the maritime industry. The company is part of the maritime industry in western Norway, and the whole production value chain is controlled by the company and concentrated in the country. The company has its own design, manufacturing, marketing, and after-sales services. The company also has a wide range of products, and for some years, it has been producing environmentally friendly products that have lower energy consumption, noise, and vibration.

### 4.3.1. Sustainability Accounting and Reporting at Company C

The managers point out that customer requirements regarding sustainability reporting vary. For instance, a customer in China may not set high expectations regarding documented environmental performance compared to a customer in Europe. Moreover, there can be different expectations from supply chain actors and the end-customer. The data collected for the social and environmental layers cannot fulfill the information needs of all the layers; thus, the company have data only on the production stage (Klymenko and Nerger, 2018).

Regarding environmental accounting, the company is currently focusing on internal reporting to improve its environmental performance. An internally developed sustainability roadmap has been distributed among the employees. The roadmap follows two main goals. The first is to increase awareness of sustainability as a topic and share the status of sustainability performance with the employees. The second is to share the developed sustainability strategy map by involving and encouraging employees—who perform different roles in various departments—in the process of improving the company's sustainability performance. Part of the report addresses the identified key goals from the 17 UN SDGs. The environmental part of the roadmap defines a strategy for more effective use of energy and materials in production. Hence, the internal road map increases the awareness of employees regarding sustainability reporting. The company states that external reporting has not been prioritized mainly because of the few requirements from the customer side. Thus, external pressure and regulations are driving forces for the company to engage and fulfil the upcoming expectations and requirements.

### 4.3.2. The Role of Digital Technologies at Company C

The company has automatized some of its manufacturing operations. However, the company is still at the planning stage when it comes to applying digital technologies to engage in more environmentally friendly, social, and ethical operations while creating

economic profit. The company is currently using sensors for the remote monitoring of mechanical equipment in vessels to detect problems and evaluate the conditions. Live monitoring of equipment is available on more than 20 vessels, which provides online data that are transferred to the control center in Norway. The managers see the growing demand for such kind of monitoring systems. Hence, they are aiming to commercialize this in future by offering a package with monitoring and service. The idea of these systems is to provide an opportunity for early detection of problems and then provide the necessary maintenance and repair on demand, which altogether can increase the lifespan of the equipment and decrease the chances of larger failures. Recently, due to the travel restrictions related to COVID-19, the company has started running extended tests at the vessel location to avoid unnecessary travel and detect the specific solution that is needed for the specific cases. A respondent suggested that they evaluate ideas on digitalized supply chains where it would be possible to monitor suppliers and their practices by collecting information regarding product materials and components. However, there is a possibility that customers may not accept interconnectedness with all suppliers. Nevertheless, they see multiple benefits of this opportunity.

> *"Many suppliers would like to be connected with the customer, but it does not always mean that the customer wants it to become dependent on the supplier, so this is a negative aspect of the digitalized value chain. On the other hand, we have an ideal cooperation with one supplier for optimizing the production processes. If you make sure that the entire supply chain is digitally traceable, you can see all the components and simultaneously plan the entire supply chain. Then, you can perform a great deal of optimization regarding waste in the value chain."* (Respondent)

Hence, the decision to digitalize the supply chain actors in the shipbuilding value chain would depend not only on a particular supplier but also on the joint decision of all value chain actors. This can hinder the digitalization along the value chain necessary for data tracking and data availability for sustainability accounting at the supply chain level. The managers state that they could optimize the existing business models and possibly introduce alternative business models that would also require technology solutions—for instance, for taking back equipment at the end of the life cycle, rebuilding, and selling it again for reuse.

### 4.4. Company D, Plastic Products Manufacturer

Company D is a family-owned manufacturer of plastic products for different industries. The company has automated production supported by internal competence and participation in research projects involving external partners. The engineers' team works closely with each customer in the product development phase to make the product competitive in the market.

### 4.4.1. Sustainability Accounting and Reporting at Company D

Sustainability represents an important dimension of this company's values and strategy. The first efforts to engage in environmental certification dated to the end of 1960. A respondent stated that the company's customers decide whether reporting on environmental issues associated with the products should be provided. Some customers require more detailed environmental impact information than others. This is a case of customers asking for detailed data on the environmental footprint that can be used for EPDs. The respondent claims that the company's ability to provide the environmental information for a product may represent a competitive advantage for the company, especially in meeting the new requirements and regulations.

> *"There will be regulations, and there will be pushes from the market, and if we can be at the forefront and have the solution ready, it will create dependence on us and create new opportunities in new markets that place greater emphasis on sustainability and responsibility for consumption and production."* (Respondent)

The company performs a life cycle analysis that demonstrates the reduction of $CO_2$ output by 75%. The company also complies with the REACH directive and ISO 14001:2015 and 9001:2015 standards.

Company D performs continuous product development aiming to improve the circulation of plastic materials. For example, they developed a new product using 100% recycled marine plastic materials.

Manufacturing is carried out in a closed system without any emission to the environment. Furthermore, the company monitors energy consumption during production. Production runs for 24 h a day, 5 days a week.

### 4.4.2. The Role of Digital Technologies at Company D

The company has a high degree of automation for its production processes, which is one of the core competences of the firm. Although the company has automated the production process, digital technology is not widely used for sustainability accounting. For instance, the monitoring of energy use in production is performed manually. Material suppliers provide data on the environmental footprint, which are stored in the internal database. Furthermore, the enterprise resource planning (ERP) system provides data on the production date of components as well as materials used in the products. The company is also involved in an initiative that evaluates opportunities for tracking plastic along its lifespan through block chain technology. The idea is at an early stage and involves different actors that can be involved in the plastic value chain comprising waste pickers, recycling actors, and manufacturers that use recycled plastic material. The digital platform can track and store information on the plastic materials from cleanup to processing.

### 5. Discussion

Previous studies within sustainability and digitalization have mostly focused on the role of digital technologies in enabling more sustainable manufacturing operations, improving production processes, and increasing efficiency [20,21]. In this study, the emphasis is on sustainability accounting and the role of digital technologies in supporting these processes.

Regarding sustainability accounting practices, we found that companies do not perform comprehensive internal and external sustainability accounting and reporting. This indicates that companies are not prepared to reveal their environmental and social performance to the public. While Company C and Company D cases underline their limited role as suppliers—which hinders opportunities for accounting along the lifespan of a product—Company C states that it is challenging to organize this type of monitoring, despite a high technology level in manufacturing. Furthermore, there is limited information shared between case companies and their suppliers, which does not allow for traceability between suppliers and a focal company. To illustrate, Company D receives only part of the material information and parameters from the suppliers. Internal strategy and own motivation are factors that allow for the collection of data for sustainability accounting and reporting, which was revealed in the example of the Company A case. In other cases, such as Company B, the main motive for sustainability accounting comes from the customers' expectations and requirements, while the findings from Company C indicate that their main motive comes from regulations and sometimes customer pressure for environmental and social practices. Findings from both Company B and Company D indicate that the availability of well-documented environmental information can be used as a sales argument for customers to choose them as a supplier. For the social layer, Company C and Company B's reporting is limited to the level of employees, including information on health and safety in production, training and educational programs, and anti-discrimination measures. Moreover, in social responsibility reporting, the companies describe the value and contribution they provide to the local community. Despite the growing tendency for environmental accounting through greenhouse gas (GHG) accounting, internal environmental certifications, and compliance according to requirements, the parameters for social layer accounting are still not well identified. At the strategic level,



the companies in this study communicate a clear ambition regarding sustainability issues. However, we observed that they tend to fulfil minimum requirements in sustainability reporting and accounting. The findings indicate that strategic decision-making is mainly supported by economic benefits; otherwise, companies only engage in sustainability to comply with regulations or respond to specific customer pressure. The absence of reporting and accounting standards for sustainability allows for a certain degree of freedom for companies in deciding what information they want to share with or keep from the public.

Regarding the role of digital technologies for sustainability accounting, the findings in this study reveal that managers do not consider the whole potential that digital technologies can bring to sustainability accounting operations. All manufacturers investigate the use digital technologies in manufacturing to a large extent; from incoming raw materials and components with stock levels in digital records in the inventory management system, and through the production process with highly digitized shop floors with extensive use of computer-controlled robots and milling, drilling, sanding, and painting machines. We have not identified any workstation that does not have a high degree of digital technologies embedded. Moreover, the stock level of finished goods is kept in digital records in inventory management systems. From this, we can see that the manufacturers are highly digitized, leading to an abundant amount of data available. However, we observe that the digitally enabled equipment are digital islands with limited or no connections between them. Some are connected vertically to management systems, but these are solely used for managing the equipment's operations. We did not identify a structured approach to harvesting the vast amount of digital data available for usage as input for economic, environmental, or social accounting. Similarly, the research of Burritt and Christ [12] highlights that some firms lack the appropriate technology for collecting data for environmental accounting.

We found some examples that illustrate how IIoT can be used to sense some of the sustainability accounting parameters and provide data needed for artificial intelligence to evaluate problem areas and develop smart solutions [62]. Additive manufacturing can improve the parameters by optimizing the use of energy and water in production, reducing waste, and improving the use of resources and materials [20]. Table 2 presents a summary of digital technologies already applied or considered for the future application by the case companies with respect to their implementation for sustainability accounting and reporting.

The table shows how digital technologies both implemented and planned to be implemented are expected to impact on sustainability accounting. Company A is more mature regarding digitalization covering manufacturing processes and the product use phase and has a higher potential in applying digital technology for environmental values. Company B sees a high potential in the digitalization of product development and manufacturing operations, but it has relatively lower positioning in terms of Industry 4.0 application for sustainability accounting. Company C and D see high potential in future applications of Industry 4.0. We see signs of the COVID-19 pandemic having been a catalyst for digitalization as it has led to an increase in the use of digital technologies across functions, including remote testing and travelling.

In general, we observe a shift from a focus on sustainability issues related to the internal manufacturing process, to a focus on sustainability issues for the entire life cycle of the product. This includes a drive towards exploring new business opportunities in the post-sales phases of a products life. Consequently, the demand for sustainability accounting at the supply chain level will increase as supported in the study by Burritt and Schaltegger [63], who underline the recognition of a new entity for accounting allowing for broader measurement and disclosure required for supply chains.

Table 2. The role of digital technology and its implementation for sustainability accounting.

| Company | Digital Technology: Implemented/in Planning Phase | How It Contributes to Sustainability Accounting and Reporting |
|---|---|---|
| Company A | Sensors for product lines—implemented. Installation of sensors is the first operation conducted for newly implemented product lines. It provides data on energy and material use in production. | Provides data that contribute to the general environmental profile with economic evaluation of a product at the manufacturing stage. |
| | Machine learning—implemented. It processes data provided through sensors and directed to the optimization of product line processes, energy efficiency in production, more efficient material use, and general optimization of a product profile. | Provides data on improvement in production efficiency, environmental performance, and economic gains that in total can present changes gained during the taken period compared to the previous ones. |
| | 3D printing—implemented. Product development and improvement of product design that can reduce defects in manufacturing. | Not directly associated with sustainability accounting and reporting. |
| | Sensors and system—implemented, provide data on internal water and energy use. | Internal reporting requires these types of data. |
| | PIM and BIM databases—implemented. Collect and structure data for digitalized value chains. All data that include all physical, mechanical, hydraulic, and environmental information on a product are saved in the PIM database. Thus, the building company can easily access complete data regarding the whole building as well as data on a specific component. | Primarily for internal accounting and reporting that contains complete data on both the whole project or building and each specific component. |
| | Digital monitoring systems for pipes—implemented. With the help of small sensors mounted inside pipes combined with NB-IoT telecommunication technology, the information on water conditions, temperature, water level, and other data will be collected and sent to a Cloud platform. This remote monitoring control providew an opportunity for customers to have an early identification of problems. | Can potentially track data for sustainability accounting and reporting. |
| Company B | Digitalization of product development—implemented. Joint digital platform based on 3D modeling accessible to employees responsible for each stage from design to final manufacturing. Reduces lead time. | Not directly associated with sustainability accounting and reporting. |
| | Robotization and automation of manufacturing processes—implemented. Make production processes more effective. Planning of operations with painting robot reduces water wastage during processes. | Not directly associated with sustainability accounting but can be equipped with tools and programs for monitoring energy and material use, amount of waste. |
| | QR-codes—in planning phase. Directed toward tracing a product's information along the lifespan supplemented by supplier information. | Can potentially store and provide data on supplier environmental and social performance. |
| | Machine learning—in planning phase. (a) Evaluates collected information on energy consumption for forecasting energy demand, and can manage energy usage, reducing it. (b) Facilitates the sorting of complex materials, separation of complex waste, improves recycling efficiency. | Machine learning monitors and collects information on energy consumption in production and recycling of materials. This information can be used for internal and external reporting. |

**Table 2.** *Cont.*

| Company | Digital Technology: Implemented/in Planning Phase | How It Contributes to Sustainability Accounting and Reporting |
|---|---|---|
| Company C | Sensors that provide data on product location and how the equipment were used during the lifespan—in planning phase. These data can be used for maintenance based on the conditions in which the equipment were used. At the end of the skip life cycle, the equipment can be detected and taken back to the producer to prolong its use through rebuilding, reusing, or recycling. | Sustainability accounting and reporting can be expanded using data on how the products are being treated during and at the end of the life cycle—for example, recyclability and reuse rates. |
| | Sensors, RFID—in planning phase. This could provide more transparent data regarding suppliers, the company's environmental performance, social, and ethical practices. The company plans to demand documentation regarding recyclability of components and detailed data on materials used. | Ability to expand sustainability accounting to the supply chain level by incorporating information regarding suppliers toward management practices at the end of the life cycle. |
| | Digitalized supply chain for maritime equipment—in planning phase. Provide an overview of all the components and materials that the product consists of, including various materials and electric and electronic components. Mapping of the components and elements can simplify the recycling process. | Ability to expand sustainability accounting to the supply chain level by incorporating information regarding product components, materials, and emissions. |
| Company D | Blockchain technology for digitalized plastic supply chain—in planning phase. Tracking and storing information on plastic from cleanup to processing. Contribute to sustainable and circular production and consumption. | Provide data on reuse and recycling of plastic. |
| | ERP systems—implemented. Allows tracking information on manufactured products. Each component is registered in the database with information on raw materials batch used for production, some of the parameters, and the date of production. | Allows tracking of basic parameters of raw materials that were used for production of components. |

The companies in the Norwegian context are well aware of sustainability issues. Moreover, the products that are made in Norway have a status in the international market that guarantees to customers that manufacturing took place in favorable working conditions for employees and that the choice of materials is based on regulations for minimizing the use of forbidden chemicals. The four cases in this study prove that sustainability plays an important role in organizational performance. Similarly, the level of digitalization and the use of technology in the case companies are high, which makes their production competitive compared to low-cost countries, which are more dependent on manual work. In fact, this study proves that technology is often used to improve the efficiency of production processes and to make operations more sustainable. Moreover, technology is not directly applied to measuring environmental and social value, which supports the statement of Burritt and Christ [12] who argue that companies are lacking appropriate technology for measuring environmental parameters. This is illustrated by the statements made from respondents from Company D and Company B that they applied a manual approach in analyzing energy consumption in production.

Regarding the TLBMC framework, the case study on its application underlines that sustainability is a data- and information-demanding area. Companies do not account for sustainability parameters across the whole product life cycle as defined in the environmental layer. The idea of the TLBMC is seen as an example of an approach for accounting for economic, environmental, and social layers of sustainability, which can further be used

for various purposes of internal and external reporting. The model does not, however, take into account how digitalization can be applied in the evaluation of the economic, environmental, and social dimensions.

The study aims to expand the research within sustainability accounting and to shed light on the contribution of digitalization in achieving sustainability for manufacturing companies. Recent studies have enabled the investigation of sustainability accounting and reporting through the lens of Industry 4.0 [22], with emphases on internal and external sustainability accounting [12], and with consideration of the supply chain level for accounting [63,64]. Previous research studies highlight the importance of digitalization for enhancing sustainability; however, there is a lack of empirical studies on the adoption of I4.0 enabling technologies for supporting environmental and social value creations. The study offers insight into how manufacturing companies address sustainability at a strategic level and sustainability accounting, but lack a comprehensive approach for environmental and social accounting at the operational and supply chain level. While digitalization is implemented in manufacturing operations, there is limited use of digital technology to provide real-time data for sustainability accounting purposes.

As seen in the example of the three companies, each of them receives varying customer expectations and requirements regarding sustainability in operations. However, the role of regulation for sustainability compliance and reporting is expected to increase in future. One of the regulations is the European Green Deal issued by the European Commission [4], which is directed to stimulate investment in sustainable economic activities and requires non-financial reporting. The first part of this regulation involves banks and other financial organizations; this will be expanded to other business actors outside the financial market by 2022–2023. Practitioners could consider adopting more sustainable accounting practices that are expected to be demanded in future. The findings reveal that even already-established regulation toward environmental requirements in public procurement may not yet be implemented. Furthermore, the companies expect new emerging business models to be developed to make the transformation toward more sustainable operations possible. Practitioners could consider new technology to support the changes in business models through vertical integration, which involves various hierarchical levels of a company, and horizontal integration, by capturing the collaboration between enterprises with resource and real-time information exchanges [65].

## 6. Conclusions

This paper presented an explorative study on digitalization and sustainability accounting from the perspective of four manufacturing firms. The companies are highly aware of the demand for more sustainable operations and products with strategies for how to engage in the transition towards sustainability. Sustainability accounting supports manufacturers in decisions on balancing sustainability benefits versus its costs. However, their efforts toward sustainability accounting are limited since environmental and social regulations often are not enforced, and since the pressure from business customers and suppliers typically satisfy minimum requirements. In terms of the environmental layer, the companies have an overview of some environmental parameters for production and some from upstream suppliers regarding materials and components. Thus, the purpose of the environmental layer accounting for the complete life cycle of a product is not done.

The findings reveal that the case companies are actively using digital technology for automation and robotization, resulting in an abundance of digital data for operations; however, these data are not applied for the accounting of environmental and social values. Hence, sustainability accounting and digitalization are seen as two separate phenomena and the intersection between them needs to be better elaborated. This represents opportunities for manufacturers and supply chain actors to utilize data from Industry 4.0 technologies to facilitate sustainability accounting, using the TLBMC tool to investigate how the companies balance sustainability benefits and costs are evolving. In particular, compared to the variety of the environmental impact, there is less attention dedicated

to social parameters. Thus, the current level of social accounting in the companies lacks parameters such as social value, scale of outreach, and social impacts in order to give a bottom line.

This study has shown how manufacturers can move towards more sustainable operations and products by utilizing already-existing data in Industry 4.0 technologies. Combining sustainability accounting and digitalization is needed to stay competitive. As future research directions, it would be interesting to investigate further the combination of the hereto separately treated phenomena of sustainability and digitalization by manufacturers. How accounting can help in quantifying sustainability benefits and costs to better manage the transition towards sustainable operations and products should be further investigated in different contexts and other geographical areas. Various accounting challenges should be studied from a sustainability perspective to gain further insight in this direction.

**Author Contributions:** Conceptualization of sustainability accounting, L.L.H.; TLBMC, O.K.; methodology, O.K.; data collection, O.K., L.L.H. and B.J.; writing—original draft preparation, O.K., L.L.H., B.J.; writing—review and editing, O.K., L.L.H., B.J.; visualization, O.K.; All authors have read and agreed to the published version of the manuscript.

**Funding:** This research received no external funding.

**Institutional Review Board Statement:** The personal data processed according to Norwegian Centre for Research Data (NSD) https://www.nsd.no/en accessed on 5 December 2020.

**Informed Consent Statement:** The personal data processed according to Norwegian Centre for Research Data (NSD) https://www.nsd.no/en accessed on 5 December 2020.

**Data Availability Statement:** Data are contained within the article.

**Conflicts of Interest:** The authors declare no conflict of interest.

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
