# Peer review of "The Enabling Role of Digital Technologies in Sustainability Accounting: Findings from Norwegian Manufacturing Companies"

_systems, doi:10.3390/systems9020033_

Round 1

Reviewer 1 Report

The paper presents an interesting and emerging topic, dealing with sustainability and Industry 4.0 technologies applied within the manufacturing industry for increasing sustainability efforts. In addition, the main focus is on sustainability accounting and the efforts of explored organizations to apply data and information obtained with and from the technologies to enable environmental and social sustainability accounting. The theoretical as well as empirical part of the paper show great effort of the authors to contribute to the body of knowledge within the still not enough researched area. 

Although the paper shows great value, it needs some revision in order to be acceptable for publication.

  1. Good proof read is needed. The English language and style are fine, but there are many sentences with wrong or misspelled words, like in lines: 22 (accounting giving insights -> accounting is giving insights), 52 (environmental -> environment), 76 (Based on an overwhelming amount of information indicate???), 83 (social an environmental -> social and environmental), 280 (related to it conditions???), 374 (on detailed and parameters???), 454 (involves emission is derived???), 471 and 474 (by-products or byproducts???), 492 (comma instead of full-stop), 529 (company is mostly stores the data), 645 (tend satisfy -> tend to satisfy), 748 (are often or limited due???). Please check entire paper for similar mistakes.
  2. Within the abstract, the (3) key finding is unclear. Please rephrase.
  3. Please check all abbreviations and titles of documents or software, explain them when they are mentioned the first time, and reference if possible (like IoT, AI, NVivo, NB-IoT, RTi, ERP).
  4. Please check reference style and reformulate all according to referencing guidelines (like Yin [2016], no page annotations, should  cited parts be italic or not and similar).
  5. Please rephrase RQ1. It is unclear what it represents. Instead of using accounting in both RQs, consider using "drive" in RQ1 instead. So the question would be what drives manufacturers to use/implement/create (or similar) economic, environmental and social values when they consider sustainable accounting.
  6. Add paragraphs / restructure discussion to emphasize the answers to the RQs. They are within , but not explicitly explained.
  7. Check table 2 - in Company A, for technology Sensors and systems, it is not stated if it is in planning or already implemented. Also, it is not clear why is the order of companies as it is (C, A, B then D?). Please explain or restructure.                  

Author Response

Dear Reviewer,

Thank you for your work and your feedback on our paper, which we believe has helped us to improve the paper. Please see the attachment with our response to your comments.

We have made some additional changes in the paper based on other comments and suggestions by Reviewers. In addition, I would like to upload two documents, one with a revised version of the paper and another document where you can see the tracking of the majority of the changes we’ve made. We also left comments in the second document pointing on the changes we’ve done. 

Best regards, Authors

Reviewer 2 Report

This paper presents a relevant theme, involving aspects the Sustainability Accounting and Digital Technologies. The contribution is significant to the advancement of knowledge; however, some points need to be better detailed for a complete understanding.

Considering it is a local/ experimental analysis, indicate the region / country in the title.

In the abstract, indicate details of the specific areas of the companies and year(s) analyzed.

The 1. Introduction is comprehensive whit a god overview of problem in context, however, needs to better describe the research hypothesis vs objective.

In methods, it is necessary to detail more. Part of the text of the 2. Background back go to Methodology (eg. 2.2. The Triple-Layered Business Model Canvas (TLBMC).

The results/discussion are correctly interpreted and very detailed. If possible, it would be interesting to have some quantitative graph or comparative figure data obtained from the companies analyzed and/or other studies.

The conclusions are extensive. Suggestion to focus on the main insights obtained from the study presented.

Author Response

Dear Reviewer,

Thank you for your work and your feedback on our paper, which we believe has helped to improve the paper. Please see the attachment with our response to your comments.

We have made some additional changes in the paper based on other comments and suggestions of Reviewers. We also left comments in the second document pointing on the changes we’ve done. Thus, I would like to upload two documents, one with a revised version of the paper and another document where you can see tracking of the majority of the changes we’ve made.

Best regards, Authors

Reviewer 3 Report

Dear Authors,

Thank you very much for the opportunity to review your very interesting paper.

The paper deals with an exciting topic that is especially relevant in today’s challenges to global warming and climate change: sustainability, accounting, and digital technologies and their role in tackling these challenges. Please find below my comments and recommendations.

Within the abstract, please re-think the phrase: “(3) Some companies are already high on sustainability efforts by their 20 maintenance of products focus, but they do report this as sustainability efforts.” Does this make sense? What should the reader take away from this statement?

Contrary to your abstract, where it seems clearer to the reader, I’m struggling to understand what you aim for with your paper as described in the introduction. The introduction should stimulate readers’ interest and lay out your argument, research gap and research question. Further, your key findings and contributions could already be included in here, also highlighting practical implications of your study. Right now it seems as if you “throw” some buzzwords at the reader, missing to link the various concepts and how and why they might be interested for your research.

Your theoretical section on sustainability accounting could be expanded. Also, it seems as some statements in here miss anchoring within the relevant literature (for example on page 3): “The data collected through sustainable accounting may have several purposes. By collecting this data, companies’ present status and maturity regarding the environmental and social values for external actors. Based on sustainable accounting data, internal decisions on operations can be optimized to move towards sustainability goals, e.g. maintenance, re-use, remanufacturing, recycling goals by reduction of material consumption, greenhouse gas emissions, and waste. Furthermore, the data provides important input in companies’ strategic process enabling decisions towards sustainable business models.”

Wouldn’t it make sense to first elaborate on the different terms/concepts before working with/linking all within the Triple Layered Business Model canvas?

Concerning your research gap you state on page 6: “There is a gap in previous studies regarding how manufacturing companies that have moved to Industry 4.0 are applying digital technologies for sustainability accounting. The present study aims at filling this gap”. Let me ask a bit provocative: So what? There is a gap and what? Maybe there is a gab because it is not interesting to analyze the topic etc.? Simply the fact, that there might a gap does not motivate the interest in exploring the gap. You would need to explain why it is interesting/necessary to analyze this gap (which should be easily possible for you given what you have included in your theoretical part before).

In general, please think about reworking this section. Do you really include and present the theoretical basis as you would need it for your analyses?

At the start of your research method, please be more specific on why this method is especially suitable for your research aim. Here, you simply present generic statements without linkage to your research situation.

Did you transcribe your interviews? Please explain / add to your methodology.

Discussion/conclusion: you have a rich trove of findings, which is really great. However, you miss on linking these findings to prior theory. References are almost absent from your discussion. What are your theoretical and practical implications? How does the Triple Layered Business Model canvas contribute? Would this canvas need to be adjusted for your research/analyses purposes?

Industry 4.0: you include a vast part on Industry 4.0 in your theory at the beginning, however, as I see it, Industry 4.0 considerations are almost absent in the back-end of your paper, e.g., within the results, discussion etc. A reader might ask in here: so what? Why including Industry 4.0 at all, and especially so prominently upfront?

Again, thank you very much for giving me the opportunity to review your paper – I hope that you find my comments helpful in further improving your paper.

Best regards,

Reviewer

Author Response

Dear Reviewer,

Thank you for your work and your feedback on our paper, which we think has helped to improve our paper. Please see the attachment with our response to your comments.

We have made some additional changes based on other comments and suggestions by Reviewers. Thus, I would like to upload two documents, one with a revised version of the paper and another document where you can see tracking of the majority of the changes we’ve made. We also left some comments in the second document pointing on the changes we’ve done. We would like to add that we’ve tried to elaborate all of your comments based on the time we received for revision. We hope the revised version is already well improved and better structured.

Best regards, Authors

Round 2

Reviewer 3 Report

Dear Authors,

Thank you very much for the opportunity to review your very interesting paper. I see that you already further improved your paper. I have only a few comments: you might consider re-drafting your introduction. Even though it reads now better than before, still, as mentioned in my prior review, it is not perfectly clear to the reader what you aim for.

Discussion/conclusion: you already expanded on this section, however, especially within the theory implications, embedding of and references to prior literature are still missing. You might reconsider working on this part.

Please also see with my prior comment on Industry 4.0 – I do not see any improvement on this: you include a vast part on Industry 4.0 in your theory at the beginning, however, as I see it, Industry 4.0 considerations are almost absent in the back-end of your paper, e.g., within the results, discussion etc. A reader might ask in here: so what? Why including Industry 4.0 at all, and especially so prominently upfront?

Again, thank you very much for giving me the opportunity to review your paper – I hope that you find my comments helpful in further improving your paper.

Best regards,

Reviewer

Author Response

Dear Reviewer,

Thank you again for your comments to our paper. We have tried to respond to your feedback and reworked the paper within the very limited timeframe. I have uploaded two documents, one with a revised version of the paper and another document where you can see the changes through “Track changes” option.

We appreciate your comments and suggestions, and we hope in the revised version we have fulfilled them and demonstrates the contributions of the paper. In addition, proofreading by a professional is carried out.

Please, see the attached file with our detailed response to your comments.

Kind regards,

authors
